# Bedside 3D Visualization of Lymphatic Vessels with a Handheld Multispectral Optoacoustic Tomography Device

**DOI:** 10.3390/jcm9030815

**Published:** 2020-03-17

**Authors:** Guido Giacalone, Takumi Yamamoto, Florence Belva, Akitatsu Hayashi

**Affiliations:** 1Department of Lymphatic Surgery, Sint-Maarten Hospital, 2800 Mechelen, Belgium; Florence.belva@emmaus.be; 2Department of Plastic and Reconstructive Surgery, National Center for Global Health and Medicine (NCGM), Tokyo 162-8655, Japan; vasko3rikov1meister918@yahoo.co.jp; 3Department of Breast Center, Kameda Medical Center, Chiba 296-8602, Japan; promise_me_now65@yahoo.co.jp

**Keywords:** lymphedema, multispectral, optoacoustic, photoacoustic, lymphography, indocyanine green, lymphatic vessel, imaging

## Abstract

Identification of lymphatics by Indocyanine Green (ICG) lymphography in patients with severe lymphedema is limited due to the overlying dermal backflow. Nor can the method detect deep and/or small vessels. Multispectral optoacoustic tomography (MSOT), a real-time three- dimensional (3D) imaging modality which allows exact spatial identification of absorbers in tissue such as blood and injected dyes can overcome these hurdles. However, MSOT with a handheld probe has not been performed yet in lymphedema patients. We conducted a pilot study in 11 patients with primary and secondary lymphedema to test whether lymphatic vessels could be detected with a handheld MSOT device. In eight patients, we could not only identify lymphatics and veins but also visualize their position and contractility. Furthermore, deep lymphatic vessels not traceable by ICG lymphography and lymphatics covered by severe dermal backflow, could be clearly identified by MSOT. In three patients, two of which had advanced stage lymphedema, only veins but no lymphatic vessels could be identified. We found that MSOT can identify and image lymphatics and veins in real-time and beyond the limits of near-infrared technology during a single bedside examination. Given its easy use and high accuracy, the handheld MSOT device is a promising tool in lymphatic surgery.

## 1. Introduction

Lymphedema, either primary or secondary, is a chronic and debilitating disorder affecting millions of persons around the world. Although compression therapy is the first choice in the treatment of lymphedema, lymphatic surgery is indicated when conservative therapy fails. While dynamic contrast magnetic resonance (MR) lymphangiography readily visualizes the central lymphatic anatomy [1], reliable and non-invasive visualization of peripheral lymphatic vessels remains challenging.

Identification of lymphatic vessels by means of indocyanine green (ICG) lymphography in patients with severe lymphedema is limited due to the overlying dermal backflow. Moreover, deep and/or small vessels cannot be detected with the current technology. Detection of lymphatic vessels in obese patients with lymphedema is even more troublesome. Although ultra-high frequency ultrasound can detect small vessels, due to the inability of coaptating ICG contrast, finding lymphatics in cases of severe lymphedema can be demanding. In addition, differentiating veins from lymphatic vessels can be difficult and depends on the operator’s experience. Such hurdles motivate the search for novel diagnostic approaches to patients with severe lymphedema particularly in view of subsequent surgical treatment.

In optoacoustic tomography a laser-induced localized thermoelastic expansion generates acoustic waves from which a 3D image is reconstructed. This emerging technology has been applied in preclinical models [2] and in humans, mainly in the field of oncology to detect tumors and metastases [3,4,5], but also in muscular and inflammatory diseases [6,7]. Blood and lymph vessels can be visualized and traced without ionizing radiation thanks to the unique light absorption spectral profile by both endogenous tissue chromophores and exogenous contrast agents including indocyanine green. Kajita et al. [8] were the first to describe lymphatic vessels up to a diameter of 0.2 mm using a 3D photoacoustic visualization system (Luxonus Inc., Kawasaki, Kanagawa, Japan). However, the need to immerse the body part in water and the size of the device’s configuration with dedicated bed [9], hamper widespread application of a promising technology.

In contrast, Multispectral Optoacoustic Tomography (MSOT) in handheld mode operating at video rates can be performed at the bedside without water immersion of the limb. In addition, the handheld MSOT device in the present work can be used to apply light at several wavelengths to optimally unmix tissue and injected chromophores, while other devices are restricted to two wavelengths [2,9]. While MSOT with a handheld scanner was found suitable for clinical imaging of major blood vessels and microvasculature in healthy persons [10] and in patients with vascular malformations [11], its use to visualize lymphatic vessels in patients with lymphedema has never been described.

We conducted a pilot study in 11 lymphedema patients in order to test whether lymphatic vessels, particularly in areas of dermal backflow, could be identified and traced by multispectral optoacoustic tomography.

## 2. Material and Methods

### 2.1. Participants

We included 11 randomly selected patients with primary and secondary unilateral lymphedema in the pilot study. Patient characteristics are presented in Table 1. Two patients (one male, one female) had primary lymphedema, seven women had secondary lymphedema following breast or gynecological cancer treatment and two men had secondary lymphedema after surgery for prostate cancer. The patients’ ages ranged from 43 to 73 years. All had lymphedema stage III or IV according to the Campisi classification [12] and underwent optoelectronic volume assessment by Perometer (Pero-System Messgeräte GmbH, Wuppertal, Germany). All patients had a confirmed diagnosis of lymphedema based on the presence of dermal backflow on lymphoscintigraphy. In one patient derivative microsurgical treatment was planned before participation.

### 2.2. Study Protocol

All examinations (ICG lymphography and MSOT) were performed by the principal examiner (GG) in an outpatient setting (Figure 1a). At 1 hour prior to the MSOT examination, indocyanine green (Verdye™) was injected subcutaneously in the interdigital space and/or in more proximal regions of the lymphedematous limb. ICG lymphography was performed with standard equipment (PDE Neo II, Hamamatsu, Japan) and patent lymphatics were marked on the skin. Dermal backflow patterns were categorised following the staging of Yamamoto et al. [13]: stardust pattern; dimly luminous, spotted fluorescent signals and diffuse pattern; the fluorescent dye was widely distributed without twinkling or identifiable spots. Next, the affected limb was scanned (distal to proximal) along the marked pattern with the handheld probe of the MSOT device (Appendix A). First we checked whether lymphatic vessels identified by both ICG lymphography and MSOT matched. We then explored the regions without visible signal of ICG (e.g., in case of deep lymphatics) and regions with a stardust or diffuse dermal backflow pattern.

The unaffected limb, not injected with fluorescent dye, was used as control (Figure 1b). Clear ultrasound gel assured acoustic coupling between probe and skin. Examiners’ and patients’ eyes were protected with laser safety glasses. Total duration of the MSOT examination was about 120 min per limb. 

In patients scheduled for lymphatic surgery it is possible to verify the accuracy and reliability of vessel imaging by MSOT. Our study included one such patient, which enabled us to verify if MSOT can pinpoint the ideal position for skin incision, and if the identified vessel is indeed lymphatic.

The study protocol has been approved by the local ethics committee of Emmaus (reference EC1742 amend 3) and all participants gave written informed consent prior to enrolment.

### 2.3. MSOT Image Acquisition and Data Analysis

The study was performed using an MSOT Acuity Echo system equipped with a 3D handheld probe (iThera Medical GmbH, Munich, Germany). The system consists of a tunable optical parametric oscillator (OPO) pumped by an Nd:YAG laser, providing 9 ns pulses with a repetition rate of up to 50 Hz between 660 and 1300 nm, with a wavelength tuning speed of 10 ms between any wavelength in the range and peak pulse energy (30 mJ at 710 nm) and a 3D hemispherical handheld detector of 8.5 cm in diameter (8 MHz center frequency, 256 transducer elements, spatial resolution of 160 μm, field of view of 15 × 15 × 20 mm, XYZ axis). MSOT signals were acquired at seven wavelengths (700, 730, 760, 780, 800, 850 and 875 nm) with a pulse repetition rate of 25 Hz, every cycle lasting approximately 280 milliseconds. Penetration depth of the field of view laser in the current set-up was limited to ~2 cm with the skin line located at about 1 mm below the top of the Z-axis. Volumes were reconstructed with direct backprojection, while linear regression was used for spectral unmixing of HbR, HbO2 and ICG and visualised in real-time using viewMSOTc software (v1.2, iThera Medical, Munich, Germany). Detailed technical descriptions of the system have been published elsewhere [14].

Due to the large size of the 3D probe, it was difficult to mark on the skin the desired location for the surgical incision preceding lympho-venous anastomosis (LVA) surgery based on the location of lymphatic and blood vessels seen on the MSOT preview screen. Therefore, a custom-made 3D printed pointer has been designed and fabricated by one of the authors (GG). 

## 3. Results

Following MSOT examination, no side effects, particularly skin damage, were observed in any of the patients.

In eight out of 11 patients, both lymphatic and blood vessels could be visualized with MSOT in handheld mode, while in three patients only blood vessels were identified (Table 1). 

In eight patients, lymphatic vessels were clearly differentiated from veins during the MSOT examination by applying a slight pressure onto the skin (Figure 2, Appendix A). Lymphatic contractility was observed in two patients (Appendix A).

Furthermore, lymphatic vessels could be identified in regions with linear pattern as well as in regions without visible fluorescence (Figure 3). 

In patients with primary (Figure 4) and secondary lower limb lymphedema, lymphatic vessels were identified in areas exhibiting a diffuse ICG pattern. 

Likewise, in the upper limb, lymphatic vessels and veins were identified by MSOT in areas of pronounced dermal backflow (Figure 5).

In patient 1, scheduled for microsurgical LVA treatment, a custom-made 3D pointer aided in marking the incision site (Figure 6). 

Indeed, vessel position based on MSOT images corresponded well with intra-operative findings (Figure 7). Consequently, an end-to-end lympho-venous anastomosis between a 0.30 mm lymphatic vessel and a 0.35 mm vein was performed. Patency of the lympho-venous anastomosis was confirmed intra-operatively by ICG lymphography (Appendix A). Lymphatic structure was confirmed by postoperative histopathologic inspection.

## 4. Discussion

In this pilot study we investigated the performance of a mobile handheld MSOT device for imaging lymphatic vessels in patients with lymphedema in view of possible microsurgical treatment.

Lymphatic vessels and their characteristics could be visualized in eight out of 11 patients. In one patient, hair at the scanned region interfered with the analysis of the images. This has been reported previously [9] and is not surprising since MSOT has also been applied to hair follicle imaging [15]. We believe therefore that this case represents a purely technical issue caused by absorption of light by the melanin present in the hair. Since shaving did not eliminate the interference, the application of depilatory creams to completely remove the hair shaft, is advisable in future studies. In the two other patients where only blood vessels were detected, structural issues or inexperience with the novel technique may have played a role. 

There were three main findings in our pilot study.

MSOT differentiates between distinct types of vessels including lymphatics

The target lymphatic vessels were visible with high contrast against the background and could be clearly differentiated from blood vessels due to spectral absorption differences between hemoglobin and indocyanine green. As previously described [16], this differentiation can be confirmed by gently pressing the 3D probe against the skin: the image of only the vein, not the lymphatic, will disappear, to reappear when pressure is lifted. Although we could also evaluate lymphatic patency by ultra-high frequency ultrasound [16], this technique is limited to a depth of 1 cm, which precludes application in more proximal segments of an affected limb and in advanced stages of lymphedema. 

MSOT detects lymphatic vessels in areas of dermal backflow

Individual lymphatic vessels were visualized even in areas of dermal backflow, where near-infrared (NIR) technology falls short. While ICG lymphography can clearly visualize patent lymphatic vessels in early-stage lymphedema (linear patterns), individual lymphatics are not identifiable in more advanced-stage lymphedema (diffuse pattern). Likewise, deep lymphatics (up to ~2 cm below the skin) and microvasculature could be identified by MSOT, whereas maximal depth with NIR technology is 1.5 cm. These features, together with the ability of identifying blood vessels at the same time, are particularly important when planning microsurgery, since lympho-venous anastomoses in areas of dermal backflow have been associated with a more favorable outcome [17].

MSOT provides images in real-time with high spatiotemporal resolution

MSOT is a valuable tool when evaluating feasibility of LVA surgery. Clear visualization of lymphatic vessels and adjacent veins, inspection of their features together with correct estimation of their proximity will help the surgeon select the ideal site for skin incision. In fact, using the custom-made 3D pointer, a LVA was successfully performed between a MSOT-identified lymphatic vessel and adjacent vein.

Despite the remarkable innovations in lymphatic imaging over the last years, none of the modalities has been able to combine the visualization of peripheral lymphatic vessels and (adjacent) blood vessels in view of required microsurgical treatment including lympho-venous anastomosis. Optoacoustic technology offers several advantages over other imaging techniques used in lymphedema patients. Optoacoustic devices do not require ionizing radiation (contrary to lymphoscintigraphy) or surgical dissection of lymphatic vessels (contrary to transpedal lymphangiography), provide spatial information (contrary to lymphoscintigraphy and ultrasound) and can detect deep vessels even with dermal backflow present (contrary to ICG lymphography). Furthermore, the duration of the MSOT examination is acceptable since the visualization of lymphatics and veins can be performed at the same time which is not possible by any other imaging method. Although lymphatic mapping by ICG lymphography may be faster, adjacent veins cannot be visualized. Lymphoscintigraphy takes up to 3 h and does not provide information on blood vessels either. 

Our pilot study describes the first results of lymphatic vessel visualization by handheld 3D multispectral optoacoustic tomography. While MSOT has been applied in cancer, vascular and inflammatory imaging [4,7,11,18], its use in other fields is still emerging [19]. In addition, MSOT has been shown to provide investigator-independent, consistent and reproducible functional soft tissue characterization [20]. Kajita et al. [8,21] have previously described the optoacoustic visualization of lymphatic vessels. Their 3D system does not provide real-time images and is limited to the use of two laser wavelengths to differentiate between lymphatic vessels containing indocyanine green from blood vessels containing hemoglobin. In contrast, our images are acquired at seven different wavelengths, which increases their differentiation power.

Furthermore, the portable MSOT device with a handheld probe used in our study is suitable for bedside measurements. Besides mobility, MSOT offers real-time visualization of identified vessels, without the need for data postprocessing. Examination with the MSOT serves as an immediate pre-surgical work-up, and the custom-made pointer will help indicate precisely the site for incision. Future expansion of the wavelength spectrum beyond 1000 nm might result in the detection of other biological tissue properties (e.g., fat in lipedema patients). In addition, extending the acquisition field of view beyond the current 2 cm depth will result in visualization of deeper structures. Finally, as it is highly likely that optoacoustic technology will become more popular in all fields of medicine, increasing the operator’s experience will add to more refined data interpretation. 

Although only 11 patients were included in this study, both primary and secondary lymphedema patients were represented, as well as varying degrees of clinical severity. Furthermore, our sample included both men and women, and both upper and lower limb edema cases. Even though our MSOT images were confirmed intra-operatively in one of our patients, our encouraging results should be replicated in larger cohorts.

## 5. Conclusions

Multispectral optoacoustic tomography is a novel, non-invasive and patient-friendly approach to the detection of lymphatic vessels in patients with lymphedema. Given that the handheld MSOT device can identify and image, in real-time and at the bedside, lymphatics and veins with high spatial resolution beyond the limits of near-infrared technology, it is a promising tool in the preoperative assessment of patients requiring lymphatic surgery.

## Figures and Tables

**Figure 1 jcm-09-00815-f001:**
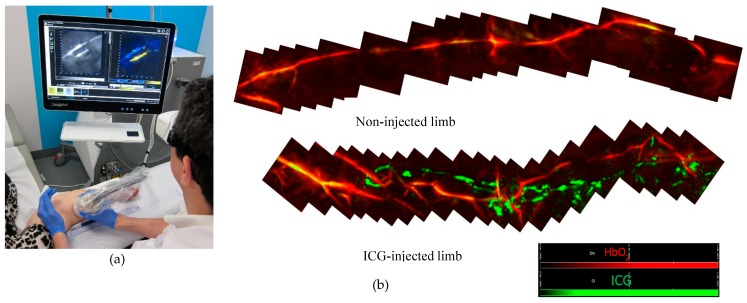
(**a**) Bedside Multispectral Optoacoustic Tomography (MSOT) examination. (**b**) In the injected limb both lymphatics (green) and blood vessels (red) were detected while in the non-injected limb, only blood vessels (red) could be detected.

**Figure 2 jcm-09-00815-f002:**
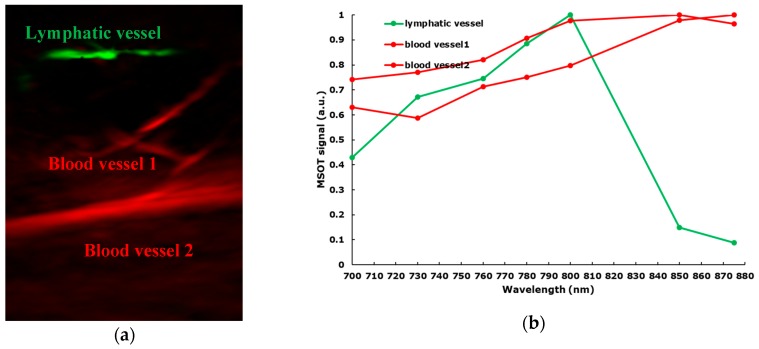
(**a**) MSOT images showing blood vessels (red) and lymph vessels (green); (**b**) the optoacoustic spectra observed in the blood and lymph vessels shown in (**a**) closely resemble the expected absorption spectra of blood and indocyanine green (ICG) respectively.

**Figure 3 jcm-09-00815-f003:**
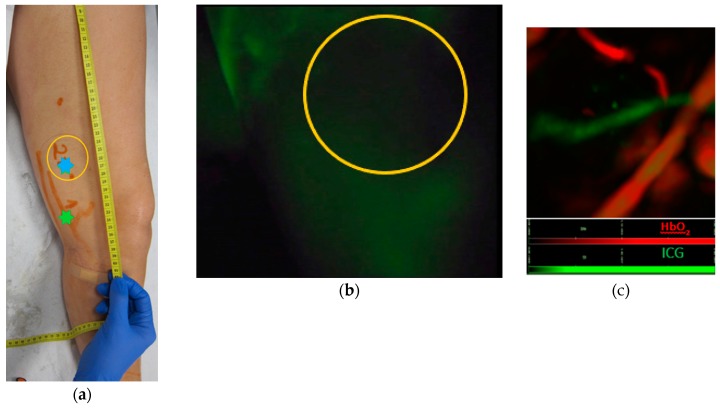
(**a**) Patent lymphatics detected by ICG lymphography are marked on the skin with an orange marker. On the linear pattern, there is a match (green star) between the lymphatic vessel detected by ICG lymphography and MSOT (image not shown). (**b**) In the area proximal to the linear pattern, no fluorescence was visualized (yellow circle) but a lymphatic vessel (blue star) was identified by MSOT. (**c**) MSOT identification of lymphatic vessel (green) and veins (red) in the area of the limb without visible fluorescence, see yellow circle with blue star in (**a**).

**Figure 4 jcm-09-00815-f004:**
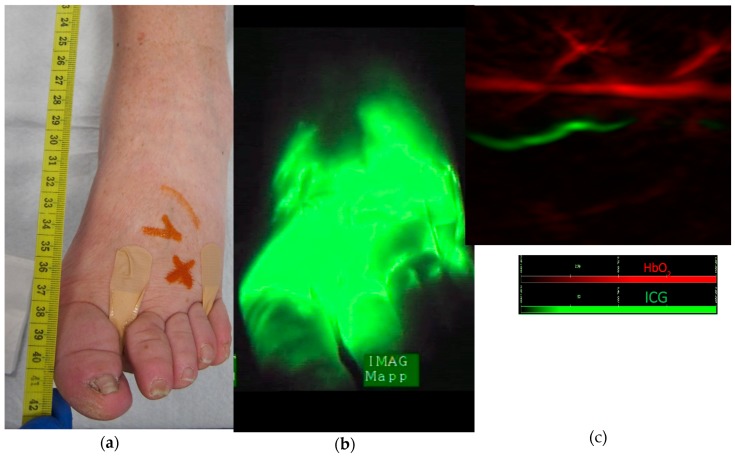
(**a**) Patient with primary lymphedema. (**b**) Diffuse pattern after ICG injection between the toes, no individual lymphatic vessel can be identified (**c**) MSOT identification of lymphatic vessel (green) and vein (red) in the area of diffuse dermal backflow. The precise localization of these identified vessels is marked on the skin with an orange cross (see Figure 4a).

**Figure 5 jcm-09-00815-f005:**
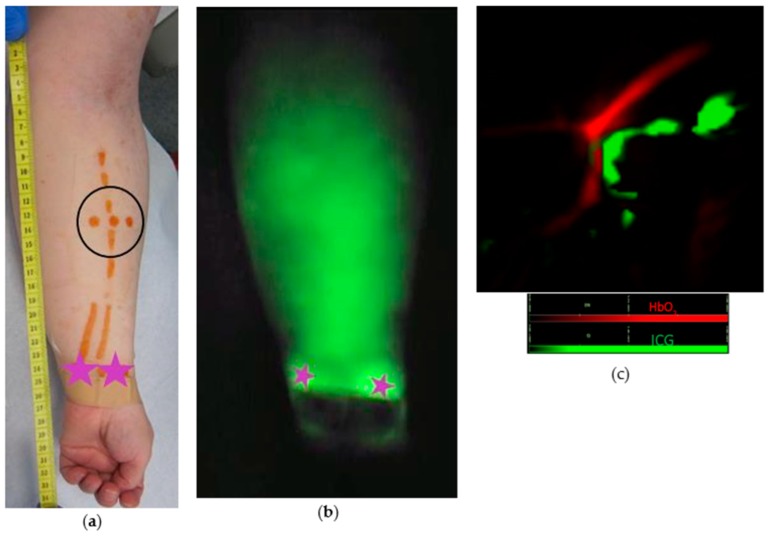
(**a**) Clinical image of patient 1 with upper limb lymphedema. (**b**) Diffuse dermal backflow pattern after ICG injection (purple stars) (**c**) MSOT identification of lymphatic vessel (green) and vein (red) in the area of diffuse dermal backflow. The location of these vessels identified by MSOT is marked on the skin by orange dots (see within black circle in a).

**Figure 6 jcm-09-00815-f006:**
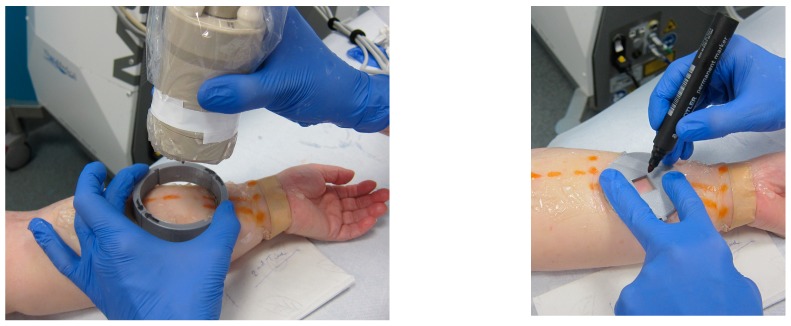
The incision site for surgery was marked with the help of a custom-made pointer.

**Figure 7 jcm-09-00815-f007:**
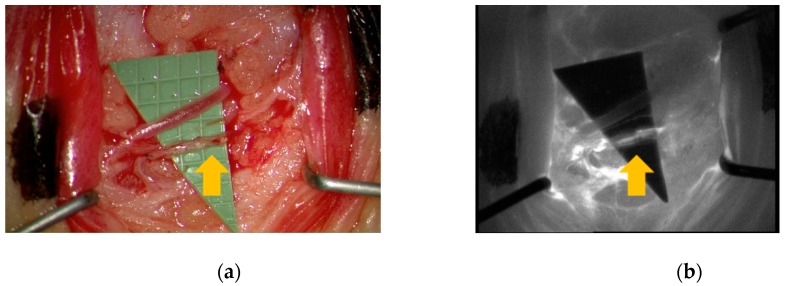
(**a**) Intra-operative visual inspection and (**b**) ICG lymphography confirmed MSOT-identified lymphatic (yellow arrow) and blood vessels.

**Table 1 jcm-09-00815-t001:** Characteristics of the patients included in the pilot study.

Patient Number	Gender	Age	Primary or Secondary Lymphedema	Prior Cancer Treatment	Localization of Lymphedema	Lymphedema Staging	ICG Patterns on Lymphography	MSOT Detection of Lymphatic Vessels	MSOT Detection of Veins
1 *	F	63	Secondary	Breast	Left arm	4	L/SD/D	Yes	Yes
2	F	72	Secondary	Breast	Right arm	4	D	No	Yes
3	M	66	Secondary	Prostate	Left leg	3	L/SD	No (hair)	Yes
4	F	49	Secondary	Cervix	Left leg	3	L/SD	Yes	Yes
5	f	74	Secondary	Breast	Right arm	4	L/D	Yes	Yes
6	M	63	Secondary	Prostate	Left leg	3	L/SD/D	Yes	Yes
7	M	73	Primary		Left leg	3	D #	Yes	Yes
8	F	43	Secondary	Cervix	Right leg	4	D	No	Yes
9	F	72	Secondary	Endometrial	Left leg	4	D	Yes	Yes
10	F	54	Secondary	Cervix	Left leg	4	L/SD/D	Yes	Yes
11	F	60	Primary		Left leg	4	D #	Yes	Yes

* Patient scheduled for lymphatic surgery; L = linear pattern, SD = stardust pattern, D = Diffuse pattern; # no migration from the injection site.

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
