# Peer review of "Bedside 3D Visualization of Lymphatic Vessels with a Handheld Multispectral Optoacoustic Tomography Device"

_jcm, 2020, doi:10.3390/jcm9030815_

Round 1

Reviewer 1 Report

The reviewer thanks the Editors for the opportunity to review this interesting manuscript. The authors describe lymphatic vessel imaging by means of MSOT, mainly intended to plan surgical interventions. 

MSOT is an emerging field of (molecular) imaging and reports on photoacoustic imaging of the lymphatic system are limited. At the same time, researchers and clinicians currently rediscover the lymphatic system for novel diagnostic and therapeutic approaches in a variety of diseases. Hence, this manusscirpt tackles an interesting field of clinical medicine and should definitely be considered for publication.

However some points should be addressed before publication:

Abstract:

-The abstract should be rewritten and restructured:

  • Line 22-26 contains over three sentences nearly the same information.
  • Line 17/18: if it is known that optoacoustic allow for exact identification of lymphatic vessels, then why this study? Rephrase.
  • Line 17: "never be imaged" this is not true. In preclinical models for e.g. see Lee (PMID: 26408999) with two wavelenght. Please rephrase here and consider this work in introduction and/or discussion
  • Line 16: "overcomes". MSOT may do better than fluorescence, but has only 3cm penetration depth. So deep tissue imaging considering techniques like MRI is not overcome by MSOT

Introduction:

  • Please add to the introduction to the broad readership of JCM: why is it important to image the Lymphatic System? Please put this in the context of current clinical challenges and research regarding the lymphatic system (for e.g. have a look at work/outviews from Maxim Itkin, Pennsylvania, US) or in your field of lymphatic surgery. 
  • Line 43: There is now a further report from this group (PMID: 31165483).
  • Please add work on lymph node/melanoma MSOT imaging (for e.g. by Stoffels et al,JAMA Netw Open, or Stoffels Sci Transl med).  Also add work on thyroid cancer (Roll et al PMID 30850507) to the former MSOT applications.

Methods

  • What does randomly selected mean in detail? From what cohort of patients? This point need clarification
  • Figure 1: Rearrange for a more appropiate design regarding size and arrangement of items. Why left panel and then A-D? Please label for e.g. as A-E (this accounts throughout the manuscript; sometimes labels a) b), then A, B, then no label). Please unify. Figure 1E is actually results, please reorder.
  • Figure 2: Figure 2 as well shows results, please reorder to next section. Also throughout the manuscript the reviewer misses the RUCT (ultrasound) images from the Acuity Echo. Please add, and also add overlay images.

Results:

  • Please explain at some point (probably best in methods) in the manuscript the patterns stardust, diffuse and linear and criterias how to categorize. Not all readers might be familiar with these patterns.

Discussion:

  • Please discuss MSOT of lymphatic vessels in the context of other imaging modalities like lymphoscintigraphy, transpedal/transnodal lymphangiography with lipiodol, MRI Lymphangiography (also work on interstitial MR-Lymphangiography Pieper/Schild Bonn, Germany PMID: 29665587).
  • Please discuss the long scanning time of your setup (120min) with regards to other methods (see above)
  • Please add, since it is a dermal application of MSOT, work on MSOT in systemic sclerosis to the former MSOT applications (Masthoff et al., PMID: 29974645).
  • Please envision on the potential use of contrast free MSOT scanning of lymphatic vessels in the future with further technical development, for e.g. by scanning at wavelenghts >1000 for fat/trigycleride detection.
  • Please check for free spaces throughout the manuscript, for e.g. line 184 or 152.
  • Please discuss the option to shave patients for a better MSOT setting.
  • Since the authors envision to use MSOT for (intraprocedural) surgical planning, they should also outline the future options to use MSOT for detection of the lymph leakage (for e.g after lymphonodectomia).

Author Response

MSOT is an emerging field of (molecular) imaging and reports on photoacoustic imaging of the lymphatic system are limited. At the same time, researchers and clinicians currently rediscover the lymphatic system for novel diagnostic and therapeutic approaches in a variety of diseases. Hence, this manusscirpt tackles an interesting field of clinical medicine and should definitely be considered for publication.

However some points should be addressed before publication:

Answer

We thank the reviewer for the encouraging words and we did our best to address all the queries.

Abstract:

-The abstract should be rewritten and restructured:

Answer

Thanky you for the remark. We have now rewritten the abstract.

  • Line 22-26 contains over three sentences nearly the same information.

Answer

We have now restructured the abstract.

  • Line 17/18: if it is known that optoacoustic allow for exact identification of lymphatic vessels, then why this study? Rephrase.

Answer

Optoacoustic technology has been previously used for the visualization of lymphatic vessels in humans by Japanese researchers. However, as mentioned further there are several differences with the device applied in this pilot study. Briefly, the iThera device uses 7 wavelengths (versus 2 for the Japanese device), does not need water immersion of the limb and above all images are reconstructed in real-time and with a handheld probe and hence applicable at the bedside of the patient allowing immediate marking of the skin in a pre-surgical setting. These differences have been described in the Introduction and Discussion section. For a better understanding of the system configuration of the device used by the Japanese researchers, we refer to figure 1a in the paper of Nagae et al., 2019. Their set-up does not allow the precise identification of lymphatics on the skin since the subject has to submerge the limb into water to be scanned (figure 1b of Nagae et al.), which is rather unpractical. As indicated in the title of the manuscript, this pilot study is the first study with a handheld probe that allows real-time identification of lymphatic vessels and veins at the bedside hence immediately before performing LVA.

Unfortunately, the maximal count for the abstract is 200 words so we were not able to address the above in the abstract, but we did in the Introduction and Discussion section.

  • Line 17: "never be imaged" this is not true. In preclinical models for e.g. see Lee (PMID: 26408999) with two wavelenght. Please rephrase here and consider this work in introduction and/or discussion

Answer

We have now paid attention to the mistake as we have rewritten the abstract.

In the study of Lee et al., 2015, no human data are presented and only two wavelengths were selected. Until today, lymphatic vessels of lymphedema patients were never imaged with 7 wavelength devices which provide a better discrimination between lymphatic vessels and veins than two wavelength devices.

Nevertheless, we have added the reference of Lee in the Introduction section.

  • Line 16: "overcomes". MSOT may do better than fluorescence, but has only 3cm penetration depth. So deep tissue imaging considering techniques like MRI is not overcome by MSOT.

Answer

Lymphatic surgery and more particularly, lympho-venous anastomosis (LVA) is performed with peripheral lymphatics. Deep tissue imaging by MRI is not required for LVA, also because MRI does not discriminate between veins and lymphatics unless special contrast agents are used. Consequently, MSOT rather than MRI can be useful in the work-up for LVA.

Introduction:

  • Please add to the introduction to the broad readership of JCM: why is it important to image the Lymphatic System? Please put this in the context of current clinical challenges and research regarding the lymphatic system (for e.g. have a look at work/outviews from Maxim Itkin, Pennsylvania, US) or in your field of lymphatic surgery. 

Answer

We have now added a paragraph regarding general aspects of lymphedema. Also, referral to the visualisation of the central lymphatic systems is made (reference 1) although visualisation of central lymphatics is not indicated for microsurgical treatment in case of peripheral secondary lymphedema. Disorders and visualisation of central lymphatics is a chapter on itself and is beyond the scope of this study.

  • Line 43: There is now a further report from this group (PMID: 31165483).

Answer

Indeed and that report was already mentioned in the text (previous reference 16) now reference 21.

  • Please add work on lymph node/melanoma MSOT imaging (for e.g. by Stoffels et al,JAMA Netw Open, or Stoffels Sci Transl med).  Also add work on thyroid cancer (Roll et al PMID 30850507) to the former MSOT applications.

Answer

We thank you for the suggestion. We have included the work of Stoffels et al. (see reference 5) but not by Roll as this work is beyond the scope of this study.

Methods

  • What does randomly selected mean in detail? From what cohort of patients? This point need clarification

Answer

All patients that visited the outpatient clinic of the principal investigator between October 2019 and December 2019 and agreed to participate in the study.

  • Figure 1: Rearrange for a more appropiate design regarding size and arrangement of items. Why left panel and then A-D? Please label for e.g. as A-E (this accounts throughout the manuscript; sometimes labels a) b), then A, B, then no label). Please unify. Figure 1E is actually results, please reorder.

Answer

Indeed, we agree that the presentation was not clear. We have now reordered the figures and unified the legends.

  • Figure 2: Figure 2 as well shows results, please reorder to next section. Also throughout the manuscript the reviewer misses the RUCT (ultrasound) images from the Acuity Echo. Please add, and also add overlay images.

Answer

We have now reordered the figures.

The device used in this study was the Acuity Echo but the ultrasound mode was not used (because does not operate at ultra-high frequencies required for visualisation of lymphatics). We only used the 3D probe. 

Results:

  • Please explain at some point (probably best in methods) in the manuscript the patterns stardust, diffuse and linear and criterias how to categorize. Not all readers might be familiar with these patterns.

Answer

The distinct patterns are now explained in the Methods section. Linear pattern: strait lines according to the anatomy of lymphatics proceeding from distal to proximal regions (axillary or inguinal nodes).

Stardust pattern: dimly luminous, spotted fluorescent signals. Diffuse pattern: the fluorescent dye is widely distributed without twinkling or identifiable spots. (see reference article by Yamamoto et al., number 13)

Discussion:

  • Please discuss MSOT of lymphatic vessels in the context of other imaging modalities like lymphoscintigraphy, transpedal/transnodal lymphangiography with lipiodol, MRI Lymphangiography (also work on interstitial MR-Lymphangiography Pieper/Schild Bonn, Germany PMID: 29665587).

Answer

We have now added the comparison to other techniques in the Discussion section.

The lymphoscintigraphy is a diagnostic tool that can not identify and precisely localize lymphatics under the skin and requires radiation. Transpedal lymphangiography with lipiodol is not performed because in order to do so you need to dissect (and interrupt irreversibly) peripheral lymphatics which would additionally cause harm since these patients already have a bad lymphatic circulation. Transnodal lymphangiography with lipiodol is not intended to visualize peripheral lymphatics, but only central lymphatics. So these 3 techniques are not applicable for the purpose, i.e. to identify (peripheral) lymphatics and veins for microsurgery.

  • Please discuss the long scanning time of your setup (120min) with regards to other methods (see above).

Answer

We have now inserted a paragraph regarding the scanning time. Furthermore, the duration of the MSOT examination is acceptable since the visualization of lymphatics and veins can be performed at the same time which is not possible by any other imaging method. Although ICG lymphography provides lymphatic imaging in real-time, adjacent veins can not be observed. Lymphoscintigraphy takes up to 3 hours and also does not provide information on adjacent veins.

  • Please add, since it is a dermal application of MSOT, work on MSOT in systemic sclerosis to the former MSOT applications (Masthoff et al., PMID: 29974645).

Answer

We believe the work on MSOT in systemic sclerosis is very interesting but beyond the scope of this study.

  • Please envision on the potential use of contrast free MSOT scanning of lymphatic vessels in the future with further technical development, for e.g. by scanning at wavelenghts >1000 for fat/trigycleride detection.

Answer

We have now speculated on this, although it is not within the scope of this study.

  • Please check for free spaces throughout the manuscript, for e.g. line 184 or 152.

Answer

Thank you for the remark. We have now removed free spaces.

  • Please discuss the option to shave patients for a better MSOT setting.

Answer

Our patients were shaved before the MSOT examination but it seems it was not sufficient for all of them. We think the best method is to use hair removing creams (depilatory creams). This information is added in the text.

  • Since the authors envision to use MSOT for (intraprocedural) surgical planning, they should also outline the future options to use MSOT for detection of the lymph leakage (for e.g after lymphonodectomia).

Answer

Hitherto, the gold standard for the diagnosis of lymphocele is lymphoscintigraphy. Since lymphoceles are often located in the groin (after lymphadenectomy) and hence rather deep, it is probably not feasible to detect leakage with MSOT. Also, ICG lymphography is usually performed in order to detect the leaking vessel(s). Anyway, it is a good suggestion for future studies (if the limited tissue depth of the MSOT device has been overcome).

Reviewer 2 Report

Very, very... interesting paper. The lymphology needs the objective and novel tools for diagnostics. I recommend only to include a few studies with optoelectronic devices for lymphedema. Examples Below:

Taradaj et al. Eur J Cancer Care (Engl). 2016 Jul;25(4):647-60.

Smykla A et al Biomed Res Int. 2013;2013:767106

Maybe, I can suggest to include some paragraphs about optoelectronic devices to introduction or discussion sections. That's why I recommended some articles. Certainly, the authors could add more to references.

Author Response

Very, very... interesting paper. The lymphology needs the objective and novel tools for diagnostics. I recommend only to include a few studies with optoelectronic devices for lymphedema. Examples Below:

Taradaj et al. Eur J Cancer Care (Engl). 2016 Jul;25(4):647-60.

Smykla A et al Biomed Res Int. 2013;2013:767106

Maybe, I can suggest to include some paragraphs about optoelectronic devices to introduction or discussion sections. That's why I recommended some articles. Certainly, the authors could add more to references.

Answer

Thank you very much for the encouraging words.

All patients received a complete work-up in order to establish the diagnosis of lymphedema. Among the examinations performed, is the optoelectronic assessment by the Perometer. But since we consider the Perometer assessment, just like the lymphoscintigraphy, as prerequisites for establishing the diagnosis of lymphedema, we did not elaborate on this.

However, we now have mentioned that all patients were assessed by the optoelectronic device Perometer (see section 2.1. Participants).

Reviewer 3 Report

The authors have performed clinical MSOT imaging to assess primary and secondary lymphedema in patients post lymphoscitigraphy. While the premise of this study is exciting and of high interest to readers, the data presented in the current form does not support any of the 3 main findings, as claimed by the author. The paper is full of overt claims which have not been verified. There is a lack of (1) clear presentation of MSOT data from different cases such as dermal backflow, and (2) data from lymphography which would enable head-to-head comparison between the two methods. 

Major Comments: 

Inclusion of lymphatics imaging data from healthy volunteers is recommended to differentiate between normal lymphatics and lymphedema.  Another approach could be to inject ICG in unaffected limb in the same patients to determine how MSOT evaluated lymphatics compare between normal and lymhedema conditions. 

Majority of the data in fig 1 shows setting or markings on the skin, which have no clear impact on the claimed outcomes of the study. Please provide separate images from ICG lymphography and MSOT in selected patients, particularly of stardust or diffuse regions and regions where the authors claim no fluorescence was seen. The data in fig 1 is not sufficient to adequately support the claims. 

The legend in fig 1 is confusing. Does figure D correspond to the stardust region marked in panel C? Please rephrase. In another example, the authors state- " MSOT was able to identify lymphatic vessels (C) in the area of dermal backflow (red circle marked on the skin)." However the accompanying figures (panel B or C) do not support this claim. Please rephrase or provide the necessary evidence. 

The authors claim " In patients scheduled for lymphatic surgery it is possible to verify the accuracy and reliability of vessel imaging by MSOT."  (line 92), under the materials and methods section. Please cite appropriate references to support this statement. IF this is a conclusion from their observations in this study, please move this statement to the Results and Discussion section. 

Please provide more details of MSOT imaging procedure. How many repetitions or cycles were performed to generate images and data for every region? How long was each patient imaged for? 

Please explain the utility of figure 2b. By providing MSOT signals at different wavelengths, the authors have provided the spectra of hemoglobin and ICG which are very well-known. A better approach would be to depict a temporal variation in the signals or signals from different stages of lymhedema obtained from different patients. 

Please include specific data in figure 3 to support the statement: Suitable lymphatic and blood vessels for lympho-venous anastomosis were identified by the handheld MSOT device. (Line 135). Providing images of the procedure does not equate or replace the actual images of lymphatic vessels obtained from MSOT. 

Fig S1 and S2 are missing from the supplementary information provided. 

It is also suggested that authors match the scores from lymphoscintigraphy with the MSOT outcomes to better demonstrate the clinical applicability and advantage of MSOT. 

MSOT has been used to image lymph nodes in healthy and cancer patients. Please include this information n the introduction with appropriate references. 

Author Response

The authors have performed clinical MSOT imaging to assess primary and secondary lymphedema in patients post lymphoscitigraphy. While the premise of this study is exciting and of high interest to readers, the data presented in the current form does not support any of the 3 main findings, as claimed by the author. The paper is full of overt claims which have not been verified. There is a lack of (1) clear presentation of MSOT data from different cases such as dermal backflow, and (2) data from lymphography which would enable head-to-head comparison between the two methods. 

Answer

Thank you for the comment. We have tried to address the several queries raised.

Major Comments: 

  • Inclusion of lymphatics imaging data from healthy volunteers is recommended to differentiate between normal lymphatics and lymphedema.  Another approach could be to inject ICG in unaffected limb in the same patients to determine how MSOT evaluated lymphatics compare between normal and lymhedema conditions. 

Answer

The aim of the study is not to compare the morphology of lymphatic vessels in healthy and lymphedema patients. Besides, this is rather useless in view of treating patients who suffer from secondary lymphedema since the surgeon has to deal with the situation as it is. The hurdle is to find lymphatic vessels that can be used for surgical treatment. Furthermore, injecting ICG in an unaffected limb solely for the purpose of imaging is never routinely performed; the patient does not provide consent for this and more importantly, it is not approved by the ethics committee from the principal investigator’s institution. To image lymphatic vessels in healthy subjects and in lymphedema patients requires a different study-set up and accordingly a proper permission from the patient and the ethics committee.

Instead, we have added the images (see Figure 1) of the non-injected limb where MSOT is able to detect veins. In the lymphedematous limb, MSOT is able to visualize both veins and lymphatics after ICG injection.

  • Majority of the data in fig 1 shows setting or markings on the skin, which have no clear impact on the claimed outcomes of the study. Please provide separate images from ICG lymphography and MSOT in selected patients, particularly of stardust or diffuse regions and regions where the authors claim no fluorescence was seen. The data in fig 1 is not sufficient to adequately support the claims. 

Answer

We have now added Figures 3 (no ICG fluorescence), 4 (diffuse pattern lower limb) and 5 (diffuse pattern upper limb).

  • The legend in fig 1 is confusing. Does figure D correspond to the stardust region marked in panel C? Please rephrase. In another example, the authors state- " MSOT was able to identify lymphatic vessels (C) in the area of dermal backflow (red circle marked on the skin)." However the accompanying figures (panel B or C) do not support this claim. Please rephrase or provide the necessary evidence. 

Answer

We agree; the figure was confusing. We have now added Figure 4 and Figure 5 which show that MSOT was able to visualise lymphatic vessels and veins in an area of diffuse dermal backflow.

  • The authors claim " In patients scheduled for lymphatic surgery it is possible to verify the accuracy and reliability of vessel imaging by MSOT."  (line 92), under the materials and methods section. Please cite appropriate references to support this statement. IF this is a conclusion from their observations in this study, please move this statement to the Results and Discussion section. 

Answer

" In patients scheduled for lymphatic surgery it is possible to verify the accuracy and reliability of vessel imaging by MSOT." This is a logical and general deduction and no reference is available since we are the first to confirm that vessels dectected by MSOT are really veins and lymphatic vessels when checked intra-operatively. This can only be confirmed if a skin incision is made. Obviously, a skin incision is only made if prior lymphatic surgery had been planned. To make a skin incision only to prove that MSOT detected vessels match with in vivo findings is clearly never authorized.

  • Please provide more details of MSOT imaging procedure. How many repetitions or cycles were performed to generate images and data for every region? How long was each patient imaged for? 

Answer

Regarding technical information we have added: The system consists of a tunable optical parametric oscillator (OPO) pumped by an Nd:YAG laser, providing 9 ns pulses with a repetition rate of up to 50 Hz between 660 and 1300 nm, with a wavelength tuning speed of 10 ms and peak pulse energy (30mJ at 710 nm).

As mentioned in the text ‘Total duration of the MSOT examination was about 120 minutes per limb.’   

  • Please explain the utility of figure 2b. By providing MSOT signals at different wavelengths, the authors have provided the spectra of hemoglobin and ICG which are very well-known. A better approach would be to depict a temporal variation in the signals or signals from different stages of lymhedema obtained from different patients. 

Answer

This analysis is only performed to prove that the identification of lymphatics and veins is correct and that there is no possibility of mistake between Hb and ICG signals.

  • Please include specific data in figure 3 to support the statement: Suitable lymphatic and blood vessels for lympho-venous anastomosis were identified by the handheld MSOT device. (Line 135). Providing images of the procedure does not equate or replace the actual images of lymphatic vessels obtained from MSOT.

Answer

We have now added figures (Figures 5, 6 and 7) to support our statements and we have adjusted the legends of the figures.

Fig S1 and S2 are missing from the supplementary information provided. 

Answer

We agree and are sorry that the supplementary figures were missing.

However, given the multitude of figures in the revised version, we have decided to omit supplementary figures and to provide supplementary videos only.

  • It is also suggested that authors match the scores from lymphoscintigraphy with the MSOT outcomes to better demonstrate the clinical applicability and advantage of MSOT. 

Answer

The lymphoscintigraphy is mainly performed for the diagnosis of lymphedema but does not provide spatio-temporal information (i.e. individual lymphatic vessels can not be visualized). Moreover, lymphoscintigraphy does not provide information on blood vessels. MSOT on the other side, is able to localise lymphatics and veins in real-time immediately before surgery. The information obtained by lymphoscintigraphy and MSOT is complementary and not stackable.

  • MSOT has been used to image lymph nodes in healthy and cancer patients. Please include this information n the introduction with appropriate references. 

Answer

References were already provided (previous reference 1,2,8 and 12) but are now updated (see reference 4,5,10,11,14,18)

Reviewer 4 Report

The Lymphatic (lymph) system is arguably one of the most important systems of the body but its importance is only now being realised. We are familiar with seeing blood vessels but not so familiar with lymph vessels. One of the problems in realising how important the lymphatic system is, has been a lack of investigatory techniques to image the lymphatic system and understand its true role in health and disease. This manuscript describes a novel method of imaging lymphatic vessels using acoustic microscopy. It is therefore an important manuscript. My only criticism is that I would like to see more evidence in the manuscript, not in supplementary videos, that the method unequivocally differentiates lymph vessels from blood vessels in vivo. There are rather too many unsubstantiated claims in the manuscript. If this can be overcome then this could be a ground-breaking publication.

Author Response

The Lymphatic (lymph) system is arguably one of the most important systems of the body but its importance is only now being realised. We are familiar with seeing blood vessels but not so familiar with lymph vessels. One of the problems in realising how important the lymphatic system is, has been a lack of investigatory techniques to image the lymphatic system and understand its true role in health and disease. This manuscript describes a novel method of imaging lymphatic vessels using acoustic microscopy. It is therefore an important manuscript. My only criticism is that I would like to see more evidence in the manuscript, not in supplementary videos, that the method unequivocally differentiates lymph vessels from blood vessels in vivo. There are rather too many unsubstantiated claims in the manuscript. If this can be overcome then this could be a ground-breaking publication.

Answer

Thank you very much for the constructive remarks.

As a proof of differentiation between lymph vessels and blood vessels, we now have inserted additional images showing only veins in the non-injected limb and veins and lymphatics in the ICG-injected limb. (see figure 1)

Also, we have added scales to show the HBO2 (veins) and ICG (lymphatics) tracers.

Reviewer 5 Report

This paper brings about very important problems concerning with the visualisation of lymphatics in lymphedema patients evaluated for microsurgery. It is also interesting that this visualisation can be obtained with the simple method at the bedside even when diffuse patterns in LSG were noted. In advanced lymphedema until now there have been some doubts about the existence of intact lymphatic vessels that can drain the tissue fluid. The results are of the great clinical benefit.

Author Response

Answer

We are very grateful to the encouraging words of the reviewer.

Round 2

Reviewer 3 Report

The authors have adequately addressed the concerns and added significant new data to improve the overall manucript.